# Risks Associated with Quality Care among Hispanic and White Populations—A Cross-Sectional Comparison Study

**DOI:** 10.3390/healthcare12020250

**Published:** 2024-01-19

**Authors:** Ching-Fang Tiffany Tzeng, Thomas Swoboda, Charles Huggins, James D’Etienne, Hao Wang

**Affiliations:** 1Department of Emergency Medicine, Baylor & Scott White All Saints Medical Center, 1400 8th Ave., Fort Worth, TX 76104, USA; 2Department of Emergency Medicine, The Valley Health System, Touro University Nevada School of Osteopathic Medicine, 657 N. Town Center Drive, Las Vegas, NV 89144, USA; 3Department of Emergency Medicine, JPS Health Network, 1500 S. Main St., Fort Worth, TX 76104, USAjdetienne@ies.healthcare (J.D.)

**Keywords:** quality care, social determinant of health, delayed care, discriminated care, patient satisfaction

## Abstract

Quality care in healthcare is a multifaceted concept that encompasses the execution of effective medical treatments and the patient’s overall experience. It involves a multitude of factors, including effectiveness, safety, timeliness, equity, and patient centeredness, which are important in shaping the healthcare landscape. This cross-sectional study used the data from the Health Information National Trends Survey 6 (HINTS 6), which collects data on various aspects of health communication and information-seeking behaviors, to investigate the factors associated with quality care among White and Hispanic populations. All adults who participated in HINTS 6 and visited healthcare service at least once in the past 12 months were included in this study. Multivariable logistic regression was used to determine the association between quality care and delay or discriminated care with the adjustment of all other sociodemographic variables. We analyzed a total of 3611 participants. Poor social determinants of health (SDOHs) (OR 0.61, CI 0.43–0.88, *p* = 0.008), delayed needed medical care (OR 0.34, CI 0.26–0.43, *p* < 0.001), and discriminated care (OR 0.29, CI 0.15–0.54, *p* < 0.001) were all negatively associated with optimal quality care. Negative SDOHs could also be positively associated with delayed care and discriminated care.

## 1. Introduction

Quality care in healthcare is a multifaceted concept that encompasses the execution of effective medical treatments and the patient’s overall experience. It involves a multitude of factors, including effectiveness, safety, timeliness, equity, and patient centeredness, which are fundamentally important in shaping the healthcare landscape [1,2]. At present, it is widely recognized that patients play an essential role in evaluating the quality of healthcare services, and the patient’s perception of quality care is considered one of the important quality metrics in patient-reported outcomes [3]. When addressed properly, quality care is consistently linked with optimal clinical outcomes, improved patient satisfaction, and increased compliance with prescribed treatment [4]. Quality care is instrumental in fostering a robust patient–provider rapport and enhancing communication, which is pivotal in delivering patient-centered care. Notably, quality care is a quality metric endorsed by prominent healthcare organizations, including the Centers for Medicare & Medicaid Services (CMS) and the Agency for Healthcare Research and Quality (AHRQ). They measure positive health outcomes through patient satisfaction [5]. Surveys such as HCAHPS (Hospital Consumer Assessment of Healthcare Providers and Systems), Press-Ganey, and NRC Health assess multiple facets contributing to patient satisfaction as a measure of quality care [6,7,8,9].

There is variation in the literature regarding the factors associated with quality care. Patient-centered communication is often recognized as a key element of quality care, while the role of social determinant of health (SDOH) and race/ethnicity remains controversial. Effective communication can lead to better treatment adherence and increased patient satisfaction [10,11,12]. For race/ethnicity, some reports have emphasized the importance of addressing health disparities and promoting equitable care, while others argue that clinical factors should be the sole focus, overlooking the impact of race and ethnicity on health outcomes [13,14,15]. While the impact of SDOHs on health outcomes is widely acknowledged, its incorporation into healthcare systems has been controversial. Some reports might prioritize addressing SDOHs, such as poverty, housing, and education, as integral to quality care, while others focus more on clinical care and treatment [16,17]. In previous studies, associations among race/ethnicity, SDOHs, and patient-centered communication had been investigated separately. SDOHs refer to environmental conditions where people reside that affect their health, functioning, and quality of life [18,19]. These factors are commonly organized in five domains: education, healthcare access and quality, economic class/stability, neighborhood, and social/community context [19]. Of the SDOHs that affect patient satisfaction, poverty is the most important factor [20]. In the U.S. in 2021, 13% earned below the federal poverty level, which was an amount estimated to be half of what is needed to afford housing, food, childcare, transportation, and health insurance [21]. Other factors that are directly or indirectly related to poverty could also affect patient satisfaction. For example, a previous study showed that the likelihood of employed individuals reporting high satisfaction to healthcare providers was over three times higher than that of unemployed individuals [10]. Similarly, individuals with a lower education level were less likely to report high satisfaction when compared with those who received high education [10], which was partially explained as salary being positively associated with individuals’ education and training level [22]. Though many studies investigated general SDOHs related to the quality of healthcare, very few studies have investigated the specific aspects of SDOHs, such as a patient’s ability to afford food or access transportation for medical care. There has been a lack of studies discussing the difference between general SDOHs and the specific components of SDOHs. Furthermore, many individuals choose to delay their needed medical care as a coping strategy. It is uncertain whether delayed needed medical care affects the overall quality of care.

In addition, factors might be indirectly related and can act as confounding factors rather than independent risk factors. For example, certain races and ethnicities, such as African American, Hispanic, or Asian, are considered minority populations [23]. Healthcare disparity has been found among minority populations [24]. Hispanic and African American patients are less likely to report excellent care when compared to White patients [25]. During the COVID-19 pandemic, Hispanic patients were also reported to have a higher rate of delaying their medical care when compared to the White patient population [26,27]. Taken together, race and ethnicity could be considered an independent risk factor affecting healthcare quality measures. However, other studies found that race and ethnicity could also act as confounding factors when a multivariable logistic regression was performed. For instance, a previous study investigated patient care experience and found that Hispanic patients reported a worse care experience than White patients in a simple two group comparison. However, when a multivariable logistic regression was performed with the adjustment of all other variables, no statistically significant difference was found between Hispanic and White patient populations, indicating that race and ethnicity were potential confounding factors [28]. On the other hand, racial and ethnic disparities in healthcare quality can be indirect. Minority populations (e.g., African American and Hispanic/Latino) tend to have poor SDOHs, face discrimination, and may choose to delay needed medical care [23]. Moreover, previous studies also reported the association between poor SDOHs and worsening healthcare quality [29]. Since minority patients are associated with poor SDOHs, and poor SDOHs are related to poor healthcare quality, being a minority patient can thus be associated with quality of care indirectly. Similarly, self-reports of discrimination can adversely affect health by triggering negative emotional reactions, followed by altered physiological reactions that increase the risk of poor health [30]. Although these associations have been reported often, few studies have investigated these factors to determine if they function independently or as confounding factors [30].

If independent factors associated with quality care can be recognized, potential interventions focused on these factors can be implemented, which may result in improved patient-centered outcomes. Meanwhile, if quality care differences exist among minority patient populations, it is essential to further investigate the factors that can potentially promote healthcare quality among such cohorts. If proved to be valid, healthcare disparities could thus be minimized. Under these circumstances, our study aims to identify the risks associated with optimal quality care and further determine racial and ethnic healthcare disparities in the quality of care between Hispanic and White populations. 

## 2. Methods

### 2.1. Study Design and Setting

This was a cross-sectional study. Our study used the data from the Health Information National Trends Survey 6 (HINTS 6), which is a national representative survey conducted by the National Cancer Institute in the United States. HINTS 6 collects data on various aspects of health communication and information-seeking behaviors, helping researchers and policymakers to better understand how the public has access to and interprets health information [31]. HINTS 6 data were collected from March to November 2022 for the entire U.S. population, and it consisted of two modes, with respondents being offered a paper survey or web option. The sampling strategy for HINTS 6 consisted of a two-stage design. In the first stage, a stratified sample of addresses was selected from a file of residential addresses. In the second stage, one adult was selected within each sampled household. The sampling frame consisted of a database of addresses used by the Marketing Systems Group (MSG) to provide random samples of addresses. Since this publicly available survey only includes participants’ de-identified information, the regional institutional review board considers such research projects as non-human research studies. 

### 2.2. Inclusion and Exclusion Criteria 

The HINTS 6 target population is civilian, non-institutionalized adults aged 18 years or older living in the United States. All adults who participated in the HINTS 6 survey and visited healthcare service at least once in the past 12 months were included in this study. Our study excluded participants who did not visit healthcare service in the past 12 months, did not report their race and ethnicity, or had missing/error information on quality care, patient-centered communication, discriminated care, and delayed medical care. Specifically, this study only focused on White and Hispanic populations to simplify our scientific evaluation. We also excluded people with missing information on other sociodemographic characteristics, such as age, sex, marital status, and SDOH information on food, housing, and transportation.

### 2.3. Outcome Measures

To understand the participants’ overall perspective of the quality care they received, our study utilized the question: “Overall, how would you rate the quality of health care you received in the past 12 months?”. If participants answered “excellent” or “very good”, we considered they received “optimal quality care”. Participants with the answers of “good”, “fair”, or “poor” were considered as having “less optimal quality care”. This quality care outcome measure has been used in many studies previously [32,33]. It measures an individual’s perception of the quality of care received from the healthcare providers and is one of the quality metrics commonly used in patient-reported outcomes [34]. 

### 2.4. Key Variables

Key variables included patient-centered communication, feeling discriminated against in medical care, delaying needed medical care, and SDOHs. 

Patient-centered communication (PCC) was evaluated using seven questions from HINTS 6: “How often did they give you the chance to ask all the health-related questions you had?”, “How often did they give the attention you needed to your feelings and emotions?”, “How often did they involve you in decisions about your health care as much as you wanted?”, “How often did they make sure you understood the things you needed to do to take care of your health?”, “How often did they explain things in a way you could understand?”, “How often did they spend enough time with you?”, and “How often did they help you deal with feelings of uncertainty about your health or health care?”. These seven questions for the PCC evaluation have been validated in the literature [35]. The answers to these questions include “Always”, “Usually”, “Sometimes”, and “Never”, and a reverse four-point Likert-scale PCC score was created (always = 4, usually = 3, sometimes = 2, and never = 1). Each item of the PCC score ranged from 1 to 4, with a total PCC score ranging from 7 to 28. We used the median PCC score as the cutoff value based on a previously reported study [36]. Participants with a PCC score > 25 were considered to have a higher perception of PCC, and those with a score below 25 were considered as a lower perception of PCC.

Regarding feeling discriminated against in medical care, our study used the question “Have you ever been treated unfairly or been discriminated against when getting medical care because of your race or ethnicity?”. If the participants answered “Yes”, they were considered as positive. Participants with the answer of “No” were considered as negative.

Regarding delaying needed medical care, our study used the question “In the past 12 months, did you delay or not get medical care you felt you needed—such as seeing a doctor, a specialist, or other health professionals?”. If participants answered “Yes”, they were considered as positive for delayed needed care. Participants with the answers of either “No, I received the medical care I felt I needed” or “I did not need any medical care in the past 12 months” were negative for delayed needed care.

There were four questions for SDOHs asking about food, housing, and transportation security. The questions included “In the past 12 months, how often were the following statements true? (1) someone in your household cut the size of meals or skipped meals because there wasn’t enough money for food, (2) someone in your household was not able to afford to eat balanced meals, (3) someone in your household was worried about being forced to move (for example, because of eviction or foreclosure), and (4) lack of reliable transportation kept someone in your household from medical appointment, work, or from getting things needed for daily living”. The answers to these four questions were: “often true”, “sometimes true”, and “never true”. Our study considered individuals with the answers of “often true” and “sometime true” to have poor SDOHs, whereas those with the answer of “never true” are considered not to have poor SDOHs. Since food, housing, and transportation play equally important roles in daily life, individuals who answered all four questions with “often true” or “sometimes true” were considered as negative SDOHs. We considered positive SDOHs if they answered “never true” to either one of these four questions. 

### 2.5. Other Variables

Other sociodemographic variables included age (18–34, 35–49, 50–64, 65–74, and 75+ years old), sex (male and female), race and ethnicity (NHW and Hispanic), marital status (single, married, and others), education level (high school or below, some college, Bachelor’s degree or above), insurance coverage (yes or no), and income level (<USD 50,000, USD 50,000–99,999, and USD 100,000+). 

### 2.6. Data Analysis

Unweighted numbers of individuals with weighted percentages were reported for all categorical variables. General sociodemographic measures and quality care were compared between White and Hispanic populations using the Rao–Scott chi-squared test. Multivariable logistic regression models were used to determine the association between quality care and the other four key variables (delayed medical care, receiving discriminated care due to race and ethnicity, patient-centered communication, and SDOHs) with the adjustment of age, sex, race and ethnicity, marital status, education level, income level, and insurance coverage. An adjusted odds ratio (AOR) with 95% confidence interval (CI) was reported. All analyses including data-merging and final analyses with replicates were performed using STATA 14.2 (College Station, TX, USA). All *p*-values are two-sided, with *p* < 0.05 considered as statistically significant. 

### 2.7. Reporting Guideline

This study’s methods and findings followed the Strengthening the Reporting of Observational Studies in Epidemiology (STROBE) guidelines. 

## 3. Results

There were 6252 participants who completed HINTS 6. We excluded 2641 participants for various reasons (see details in Figure 1). A total of 3611 participants were considered in the final analysis, as shown in Figure 1. 

The baseline characteristics of White and Hispanic participants are shown in Table 1. In brief, Hispanic individuals were younger than White ones (White: 19.67% of individuals aged 18–34 years and 11.06% of individuals aged 75+ years; Hispanic: 35.89% of individuals aged 18–34 years and only 4.26% of individuals aged 75+ years, *p <* 0.001). More White individuals received a higher level of education (Bachelor’s degree and above, 37% vs. 22%, *p <* 0.001), had a higher income (>USD 100,000, 38% vs. 22%, *p <* 0.001), and had health insurance coverage (95% vs. 84%, *p <* 0.001) than Hispanic individuals. More White individuals were married, whereas more Hispanic individuals were single (*p =* 0.0004). The findings in Table 1 indicate the different sociodemographic characteristics between White and Hispanic individuals.

Quality care, discriminated care, delayed care, and SDOHs were compared between the Hispanic and White participants. When a simple comparison was performed, more White individuals reported optimal quality care in comparison to Hispanic individuals (*p* = 0.0001). Negative SDOHs, having unfair or discriminated care, and delaying or not receiving the medical care needed had a significantly lower rate of optimal quality care when compared to their counterparts, while individuals with positive patient-centered communication had a significantly higher rate of optimal quality care when compared to individuals with negative patient-centered communication (all *p* ≤ 0.0001), as shown in Table 2. 

We then performed multivariable logistic regressions to determine the associations among quality care, SDOHs, discriminated care, and delayed care after controlling for all other demographic variables, including age, sex, marital status, race and ethnicity, education level, income level, and insurance coverage. Negative SDOHs (AOR 0.61, CI 0.43–0.88, *p* = 0.008), delayed needed medical care (AOR 0.34, CI 0.26–0.43, *p <* 0.001), and discriminated care (AOR 0.29, CI 0.15–0.54, *p* < 0.001) were all negatively associated with optimal quality care, as shown in Figure 2. Additionally, we further determined the three-way associations among SDOHs, discriminated care, and delayed care (i.e., SDOHs and discriminated care, SDOHs and delayed care, and delayed care and discriminated care) with the adjustments of all other demographic factors. We found that negative SDOHs and delayed care as well as negative SDOHs and discriminated care were significantly associated with each other independently, while delayed care and discriminated care had no significant association with each other (Figure 2). Detailed findings of the multivariable logistic regression analyses are reported in Appendix A Table A1.

## 4. Discussion

Our study found that negative SDOHs, delayed needed medical care, and discriminated care were all negatively associated with optimal quality care. In addition, negative SDOHs were also associated with delayed medical care and discriminated care. Similar findings were found in previous studies [37,38]. Cumulative exposure to social and economic disadvantages can impact medical care outcomes in a dose-dependent fashion [39]. Therefore, interventions to improve SDOHs that allow individuals to afford meals, housing, and transportation and access the necessary medical care are essential and lead to improved quality care. On the other hand, race and ethnicity are not independent risks for reduced quality care after controlling for individuals’ SDOHs, delayed care, discriminated care, and other sociodemographic factors. Our study emphasized the differences of investigating SDOHs in detail (i.e., general income, education, and insurance coverage) versus overall SDOHs, suggesting a focus on detailed items may be needed for SDOH research [40,41].

In this study, we only chose White and Hispanic individuals for quality metrics’ comparisons. We chose these two populations based on two reasons: (1) The Hispanic population, though still considered a minority population in the US, is one of the three main populations in the country at present. According to the U.S. Census Bureau’s recent report, there are approximately 58.9% of non-Hispanic White, 13.6% of African American, and 19.1% of Hispanic/Latino population. Therefore, the Hispanic population is one of the three main racial and ethnic populations in the United States [42]. (2) Among these three populations, over half (51.1%) of the total US population growth came from the growth in the Hispanic/Latino population, which is the most fast growing population in the United States [43]. To further determine the healthcare quality differences between a rapidly growth population (i.e., Hispanic) and its counterpart (i.e., White), we selected White and Hispanic populations in this study. 

The factors associated with quality care vary across reports. Some factors are commonly considered, while others are less studied, such as detailed factors of SDOHs, delayed medical care, and discriminated care related to race and ethnicity. SDOH factors, including food, housing, and transportation, remain the most important ones [44]. These items differ from general SDOHs in that some individuals may fall outside the federal poverty levels but have food, housing, or transportation challenges [45]. For example, in 2022, 11.5% of the US population were below the federal poverty line, and 12.8% of the US population reported food insecurity [46,47]. Social determinants operate on a gradient, where each small improvement and step upward can increase the probability of better health outcomes [48]. The education level also plays a role in individuals’ income. Individuals with a high school education level or below are less likely to secure a high-income job position, have less chances of being promoted, and have a higher risk of being unemployed due to economic recession [49]. These people with a lower education level also have problems of health literacy, which could subsequently affect patient–provider communication and eventually influence individuals’ health outcomes [50,51]. Food, housing, and transportation are necessary expenses, and if lacking, it would indicate a strong financial challenge among individuals that may impact their health and wellbeing [45,52]. Therefore, detailed components of SDOHs can better reflect an individual’s living conditions, and accurate personal health and living conditions can further improve the health quality care predictions.

Due to economic inflation after the COVID-19 pandemic, some people did not access or delayed needed medical care as one of their coping strategies [53,54]. This can be associated with poor quality care due to health deterioration, medical noncompliance, or other stresses related to medical care [55]. Age, health status, delayed care, employment status, and racial discrimination all contributed to increased health burdens [56]. During the COVID-19 pandemic, those who experienced discrimination in healthcare were 304% more likely to report mental health issues, and those who lost their jobs were 56% more likely to report mental health problems in California [56]. Under these circumstances, optimal quality care is hard to achieve. The individuals’ perception of discriminated care related to race and ethnicity can affect patient–provider rapport and impair patient-centered communication, resulting in poor quality care. Previous studies have shown such associations [57,58,59]. Our study confirmed the negative association between discriminated care and quality care. Additionally, we found a positive association among negative SDOHs, delayed care, and discriminated care. Interventions to improve one factor could improve the others. Unfortunately, this study can only show the association without understanding their causative effects.

There are strengths to this study. First, our study used a national representative sampling design with a large-scale, weighted population of over 153 million patients. We further analyzed the three risk factors, including SDOHs, discriminated care, and delayed care, along with their associations (Figure 2). Our study also compared general SDOHs with detailed aspects of SDOHs, which has been rarely reported in previous studies. Lastly, our study used multivariate logistic regression with the adjustment of other sociodemographic factors to avoid confounding factors. Our findings identified three independent factors (SDOHs, delayed care, and discriminated care) as being negatively associated with optimal quality care. Our study’s findings provide evidence of SDOHs being associated with different quality metrics (quality care, delayed care, and discriminated care), specifically among the Hispanic participants. These findings can serve as a foundation to further implement potential interventions to minimize quality care disparities due to racial- and ethnic-related factors. Tolentino et al. highlighted several recommendations to improve health equity, such as conducting individual and institutional reflection and analyses of racial inequities [44].

Our study also has limitations. Individuals with missing, incorrect, or error-prone information were excluded from the study. Although this only accounted for a small percentage of the entire cohort, their exclusion may have resulted in selection bias, which might happen when the sample may not be representative of the entire population. Second, our study only investigated quality care between White and Hispanic individuals, and other races and ethnicities were not analyzed. In this study, White individuals accounted for over 80% of the entire sample size, whereas Hispanic individuals only accounted for less than 20%, indicating an imbalanced population comparison. However, this national representative survey used replicate weights on each participant that mimicked the US population distribution, thus minimizing the imbalanced population selection bias. In addition, the study might not be able to be applied to populations residing outside of the United States or non-English- or Spanish-speaking populations since these participants were not involved in this study. Also, our study only analyzed variables that existed in the survey. Other potential factors associated with quality care were not included in our analysis. Our study only determined the associations among different risk factors, instead of the causative effects. Lastly, SDOHs were subdivided into food, housing, and transportation. Other factors, such as job opportunities, physical activity opportunities, neighborhood conditions, and health literacy skills, were not assessed. Future large-scale research investigating more SDOH details related to healthcare is warranted for external validation.

## 5. Conclusions

This study found that negative SDOHs, delayed care, and discriminated care were all negatively associated with optimal quality care. Meanwhile, poor SDOHs were positively associated with delayed care and discriminated care. This is the first study comparing detailed aspects of SDOHs with general SDOHs. Future research investigating the roles of individual SDOHs in improving quality care are warranted for healthcare purposes.

## Figures and Tables

**Figure 1 healthcare-12-00250-f001:**
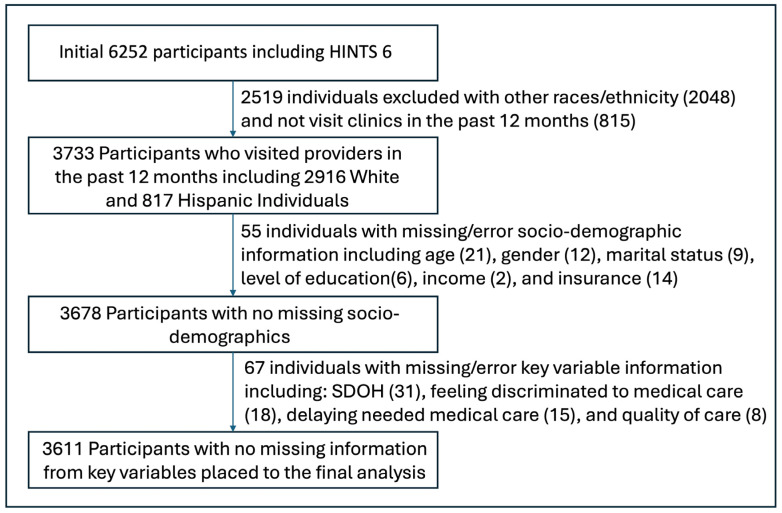
Study flow diagram.

**Figure 2 healthcare-12-00250-f002:**
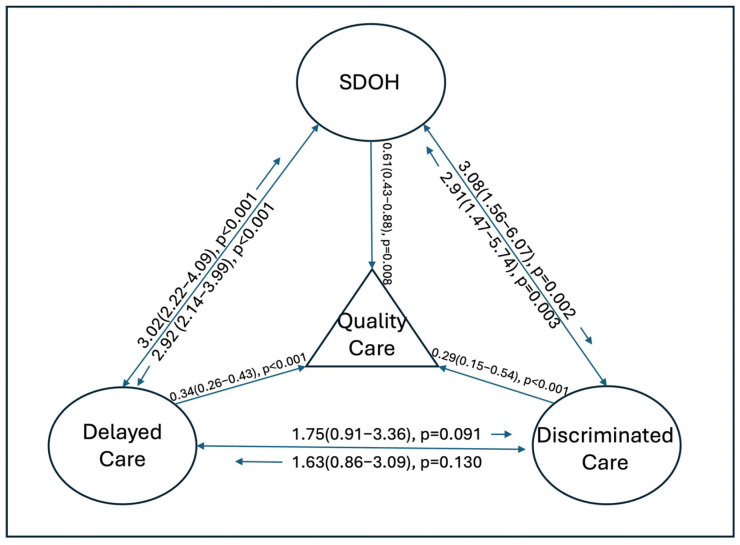
Associations of quality care, SDOHs, delayed care, and discriminated care.

**Table 1 healthcare-12-00250-t001:** Participant characteristics of White and Hispanic groups.

	White	Hispanic	*p*-Value
Number of participants—N (Wt%)	2833 (81.73)	778 (18.27)	
Age—N (Wt%)			<0.001
18–34	313 (19.67)	184 (35.89)	
35–49	495 (22.89)	194 (28.14)	
50–64	816 (30.46)	208 (22.84)	
65–74	713 (15.92)	126 (8.87)	
75+	496 (11.06)	66 (4.26)	
Gender			0.1965
Male	1149 (47.74)	274 (44.29)	
Female	1684 (52.26)	504 (55.71)	
Marital Status			0.0004
Single	397 (24.18)	155 (33.07)	
Married	1629 (62.67)	424 (55.85)	
Others	807 (13.15)	199 (11.07)	
Level of Education			<0.0001
High school or below	751 (32.16)	332 (51.65)	
Some college	558 (30.40)	162 (26.42)	
Bachelor’s degree or above	1524 (37.44)	284 (21.93)	
Income levels			<0.0001
<USD 50,000	969 (30.33)	411 (45.35)	
USD 50,000–99,999	909 (32.14)	214 (33.11)	
≥USD 100,000	955 (37.53)	153 (21.55)	
Insurance			<0.0001
Yes	2727 (95.15)	672 (84.36)	
No	106 (4.85)	106 (15.64)	

Note: Abbreviations: N, Number; Wt, Weighted.

**Table 2 healthcare-12-00250-t002:** Factors associated with quality care.

	Optimal Quality Care	Less Optimal Quality Care	*p*-Value
Race/ethnicity—N (Wt%)			0.0001
White	2136 (74.19)	697 (25.81)	
Hispanic	477 (60.00)	301 (40.00)	
Patient-centered communication—N (Wt%)			<0.0001
Positive	1611 (61.64)	173 (15.55)	
Negative	1002 (38.36)	825 (84.45)	
SDOHs—N (Wt%)			<0.0001
Negative	404 (52.39)	317 (47.61)	
Positive	2209 (76.67)	681 (23.33)	
Having unfair or discriminated care—N (Wt%)			<0.0001
Yes	56 (33.34)	92 (66.66)	
No	2557 (73.04)	906 (26.96)	
Delaying or not receiving medical care needed—N (Wt%)			<0.0001
Yes	617 (54.04)	494 (45.96)	
No	1996 (79.61)	504 (20.39)	

## Data Availability

The study data are publicly available at https://hints.cancer.gov/data, accessed on 30 November 2023.

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
