# Peer review of "Risks Associated with Quality Care among Hispanic and White Populations—A Cross-Sectional Comparison Study"

_healthcare, 2024, doi:10.3390/healthcare12020250_

Round 1

Reviewer 1 Report

Comments and Suggestions for Authors

Dear authors,

Your article addresses a fundamental topic in healthcare services.

They aim to comprehensively explore and understand the multifaceted nature of quality care in healthcare, considering various factors such as SDOH, patient-centred communication, discrimination, and delayed medical care. They also seek to identify potential disparities between different racial and ethnic groups.

However, to further strengthen the research and improve the article, the authors may consider:

 In Methodology:

While the section mentions that the study design is cross-sectional and the data source is HINTS 6, it could benefit from providing more details on the survey questions used to assess quality care, patient-centred communication, and other key variables. Consider including a brief overview of the HINTS 6 survey to give readers context about the data source.

For example, the article mentions using the "Overall, how would you rate the quality of health care you received in the past 12 months?" to measure quality of care. Please briefly explain the rationale behind choosing specific questions and how they align with the study's objectives.

Regarding the sample, the provided text does not explicitly mention it. The authors should ensure that they have adequately justified the sample size based on the study objectives, expected effect sizes, and statistical power considerations. If this information is not present in the manuscript, including it or explicitly stating why a specific sample was used would be beneficial.

Discussion

They need to expand the discussion (engage more deeply with existing literature. Discuss how the study's findings align with or differ from previous research in the field. This can provide context for the significance of the current study. 

Implications: Discuss potential policy implications of the study findings. How might the identified associations inform healthcare policies or interventions to improve quality care, particularly for populations with negative SDOH, experiences of discrimination, or delayed medical care?

Considering Long-Term Impact:

Given the potential influence of the COVID-19 pandemic on healthcare behaviours, discuss whether the study considered or controlled for pandemic-related factors and their impact on delayed medical care.

Discussion of Limitations:

Provide a more detailed discussion of the limitations of the study. For instance, discuss potential biases introduced by excluding participants with missing or incorrect information and acknowledge any limitations in the generalizability of the findings.

 Conclusion:

Reiterate the study's main contributions and explicitly state how the findings contribute to the existing body of knowledge.

Consider addressing potential future research directions in more detail.

Reviewer 2 Report

Comments and Suggestions for Authors

There are some technical issues regarding the journals' guidelines. In particular, the references in the text and at the end are not in accordance with the guidelines.

Ten references are old (before 2017). Try to replace them with more up-to-date ones.

The first part of the introduction is weak. A better presentation of quality indicators of health care services is needed. Also, the literature you use in this section regarding quality indicators is about very specific groups of patients such as tuberculosis. You could possibly present fundamental articles regarding service quality and its dimensions.

In the methodology you state that the study included 12 months of data, but you do not specify the exact calendar period (e.g. 1 July 2022 to 30 June 2023).

The discussion does not adequately support your results. Needs improvement and targeted presentation of studies compared to results. Indicate the period of COVID-19 and its impact on health status. You could link the anti-COVID-19 measures (e.g. Quarantine) and the health effects of the disease to the loss of thousands of jobs, loss of income, loss of insurance coverage, etc., and the correlation of all of these to the health status of the population. You could perhaps see if there are studies showing the impact of these factors in the COVID-19 period on Hispanic and White Populations and if there are differences between population groups.

Reviewer 3 Report

Comments and Suggestions for Authors

I think this is a very good study and analysis using large data sets.

 However, please consider the following points.

Lines 49-49, Are these references (9-11) to the paper describing the role of social determinants of health (SDOH)?

Please use more appropriate references.

In Table 1 (lines 171-174), the authors state that there are significant differences between Whites and Hispanics. However, there is no analysis of how these differences affected the results of the logistic regression analysis.

Lines 155-157 and Lines 191-192. 

Lines 155-157 and lines 191-192

"with adjustment for all other demographic factors such as age, sex, marital status, education level, income level, and health insurance coverage."

How were these variables adjusted?

I believe that the results of the above analysis will have a significant impact on our discussion of the results.

The following are minor adjustments.

Line 33 .... .... shaping the healthcare landscape.(1) 

The MDPI should be written as follows

... ... shaping the healthcare landscape (1).

Please correct others as well.

I hope this helps.

Reviewer 4 Report

Comments and Suggestions for Authors

Dear Authors:

Thank you for allowing me to review your manuscript, which deals with a topic that is important and relevant. I consider that your manuscript follows an adequate methodology and can be published, but I would like to make some comments in order to clarify some questions that require explanation.

The Introduction is adequate, although it could be improved. This study has been carried out in a specific country (I understand the USA), where there is a diversity in terms of ethnicities and important cultures. Given the subject of this work, perhaps explaining a little more the context in the country where this study has been carried out would have been desirable (explaining the distribution of ethnicities in the country, demographic data, etc.). I think this would help to understand the importance of studying this topic.

In methods there are important aspects that should be clarified.

The time period in which the study was carried out should be stated, since it is a cross-sectional study and this aspect should be indicated as indicated in the STROBE standards.

-Another important aspect is to explain where the study has been carried out. Are these data from the entire U.S. population or data from a specific region? From the sample used, it seems that the data is from a specific city, region or state. If on the contrary they are data of the whole country, where what has been collected is a representative sample, you must explain everything referred to the type of sampling carried out, how randomization has been carried out in the case of being carried out as well as the sample calculation that ensures that the sample is representative of the population under study.

-The subsections reporting guidelines and conflict of interest should be separate from the body of the manuscript (consult the editor).

-All scores obtained with respect to PCC (perception of Patient-Centered Communication) do not appear in the results. These are data that should be presented.

-One aspect that does not invalidate the study but would have increased the value of the study is if other ethnicities or races had not been excluded. I understand that this is another study. Why was this decision made? Perhaps an explanation would be appropriate in the discussion. It could perhaps be considered a limitation.

I hope my comments are helpful. Best regards

Round 2

Reviewer 2 Report

Comments and Suggestions for Authors

Considerable effort was made and the quality of the article was improved. 

Author Response

Thank you for the suggestions.

Reviewer 3 Report

Comments and Suggestions for Authors

The manuscript has been well revised. However, the following points need improvement.

Lines 72-73: "Although many studies have examined general SDOH..." This sentence needs some references to previous work on social determinants of health related to quality of care.

Line 274: The authors mention that "We chose these two populations for two reasons: 1) Hispanic population..."

The first reason is stated as Hispanic population, but where is the second reason stated?

Line 276: "According to the U.S. Census Bureau's most recent report," There is no mention of the source here.

I would appreciate it if you could take this information into consideration.
